# Training and Resources Related to the Administration of Sedation by Nurses During Digestive Endoscopy: A Cross-Sectional Study

**DOI:** 10.3390/healthcare12202087

**Published:** 2024-10-19

**Authors:** Miriam Hidalgo-Cabanillas, José Alberto Laredo-Aguilera, Ángel López-Fernández-Roldán, Rosa María Molina-Madueño, Pedro Manuel Rodriguez-Muñoz, Carlos Navarrete-Tejero, Ángel López-González, Joseba Rabanales-Sotos, Juan Manuel Carmona-Torres

**Affiliations:** 1Hospital Universitario de Toledo, 45004 Toledo, Spain; miriam.hidalgo@alu.uclm.es (M.H.-C.); angel.lopezfernandez@uclm.es (Á.L.-F.-R.); 2Facultad de Fisioterapia y Enfermería, Universidad de Castilla-La Mancha, 45071 Toledo, Spain; rosam.molina@uclm.es (R.M.M.-M.); pedrom.rodriguez@uclm.es (P.M.R.-M.); carlos.navarrete@uclm.es (C.N.-T.); juanmanuel.carmona@uclm.es (J.M.C.-T.); 3Grupo de Investigación Multidisciplinar en Cuidados (IMCU), Universidad de Castilla-La Mancha, 45071 Toledo, Spain; 4Instituto de Investigación Sanitaria de Castilla-La Mancha (IDISCAM), 45004 Toledo, Spain; 5Hospital Universitario Rey Juan Carlos, 28933 Mostoles, Spain; 6Facultad de Enfermería, Universidad de Castilla-La Mancha, Campus Universitario s/n, 02071 Albacete, Spain; angel.lopez@uclm.es (Á.L.-G.); joseba.rabanales@uclm.es (J.R.-S.); 7Grupo de Actividades Preventivas en el ámbito Universitario de Ciencias de la Salud (GAP-CS), Universidad de Castilla-La Mancha, 02071 Albacete, Spain

**Keywords:** nurse anesthetists, endoscopy, nursing, sedation, endoscopy digestive system

## Abstract

Background: The healthcare professional who performs sedation for digestive endoscopy procedures is usually the nurse. Therefore, knowledge and training on the part of the nurse is an important factor for the correct sedation of the patient and may affect, among other factors, the quality of health care and the recovery and well-being of the patient. Objective: To determine the training opportunities and resources available to the specialists involved in digestive endoscopy services in hospitals in Spain in which sedation is performed by the nursing staff. Methods: This was a descriptive cross-sectional study performed in the Digestive Endoscopy Service at Toledo University Hospital and nearby hospitals in central Spain. The sample consisted of 75 nurses who administer sedation in digestive endoscopic procedures. Results: Regarding the training of nurses, the vast majority were generalist nurses, and only a small percentage obtained specialized training through ongoing training. Most had been working in the service for more than two years, and very few had previous experience in sedation, although the vast majority currently applied sedation. Life support training was also critical. There were significant differences in the availability of resources between different hospitals. Conclusions: The training of nurses who perform sedation in digestive endoscopy services could be improved since there is a high proportion of personnel without specific training in sedation before starting to work in the service. It is crucial to implement targeted and ongoing training programs to improve competencies in this area, as they are essential to ensure the safety and effectiveness of the procedure. Given the variability in resources and personnel training that exists between different hospitals, it seems important to establish standards at the institutional level.

## 1. Introduction

Sedation in digestive endoscopic procedures, such as gastroscopies, colonoscopies, and echo endoscopies, is a crucial aspect of the procedures because it improves the patient’s tolerance of the method, minimizes discomfort, and decreases pain [1]. Several studies have highlighted the importance of the knowledge, skills, and attitudes of nursing staff in the administration and control of sedation to ensure the safety and effectiveness of the procedure [2,3].

Throughout history, sedation has been administered by medical personnel, but owing to the increasing demand for endoscopic procedures, the nurse has become a more relevant role in this context [4]. Faced with this growing demand, the need for specific training has been discussed so that nurses who perform this type of procedure can acquire the necessary skills [5]. In fact, in some countries, there is a nurse specialist in anesthesia who is trained to administer anesthesia in various clinical settings [6]. In terms of their roles and responsibilities, these professionals administer anesthesia during surgical procedures, assist in pain management, and collaborate closely with other healthcare professionals to ensure the safety and well-being of the patient. This role includes pre-procedure assessment, during-procedure assessment, pain management, and monitoring of post-anesthetic recovery [6].

The interventions used by nurses for sedation in digestive endoscopies range from the previous evaluation of the patient to the effective control of pain during the procedure and the management of possible complications [7,8].

Some authors emphasize the need to implement educational programs on sedation and resuscitation, adequately monitor oxygenation and ventilation, develop manuals and simulations to guide nurses in response to sudden changes and improve the training and knowledge of the staff on sedation and the assessment of sedation levels [9]. In fact, the current consensus in Spain establishes that specific training is necessary for nurses to be trained since there is currently no specialty of anesthesia nurses [9].

Training and specific training in sedation are essential since nursing personnel play crucial roles in the administration of sedatives and the monitoring of patients [10]. Nursing prescriptions in Spain are regulated and do not include the drugs necessary for sedation. Training is not regulated either, which is a pending issue, as there are criteria established by the International Council of Nurses and the International Federation of Nurse Anesthetists [10,11].

Digestive endoscopic procedures in Spain are based on the recommendations of the Spanish Association of Gastroenterology (AEG), the Spanish Society of Digestive Endoscopy (SEED), the European Society of Gastrointestinal Endoscopy (ESGE), the Consensus of Asia and the Pacific, the Spanish Society of Preventive Medicine, the European Panel on the Adequacy of Gastrointestinal Endoscopy (EPAGE) and the guidelines of the Spanish Ministry of Health. These establish a reevaluation of endoscopic indications, prioritize the most beneficial procedures, maintain social distancing and the use of patient masks, train nursing staff regarding procedure guidelines, reduce the volume of elective activity, and follow the times recommended by the guidelines of current clinical practice [11].

In the national context, the need to update and unify the laws that regulate these practices is increasingly evident [12]. Clinical practice guidelines and consensus documents from scientific societies and health authorities ensure that nurses can perform this procedure in specific situations, which depend on the type of test and the required level of sedation, provided they have undergone appropriate training [13].

Recent studies have highlighted the variability in the training of nursing personnel in Europe, suggesting the need to standardize training programs to guarantee safe and effective practices in all medical centers [14]. At the international level, the WHO guidelines emphasize the importance of the competence of nursing personnel in the administration of sedation, promoting the adoption of uniform training standards [15].

In addition, practical experience and participation in advanced training courses are decisive for the development of competent skills in sedation, as shown by several studies that link direct experience with significant improvements in the competence and confidence of the nursing staff [16].

An indicator of the quality and safety of the endoscopic procedure is the percentage of complications that are associated with sedation administered by nurses. While serious complications are infrequent, they necessitate a high level of preparedness and swift action from the nurses involved in the procedure [17]. Such complications may include cardiopulmonary risks, such as hypoventilation, respiratory depression, apnea, hypotension, and bradycardia [3]. In this context, sedative drugs can induce muscle relaxation, which may hinder the ability to maintain airway patency, whereas analgesics, particularly opioids, can decrease the respiratory rate. If not promptly identified and addressed, these complications can lead to hypoxemia, potentially resulting in irreversible damage to vital organs or even death [18].

In fact, continuous evaluation and evidence-based training are essential to maintain and improve nursing skills in sedation, thus ensuring the safety and well-being of patients during endoscopic procedures [19].

As mentioned above, specialized training in sedation for nurses is essential and can influence the incidence of complications. In fact, according to the advice of the main Spanish and international scientific societies, prior training is necessary for the application of sedation in digestive endoscopy services. However, studies carried out in Spain to analyze the knowledge and practices of digestive endoscopy nurses are scarce. Therefore, the main objective of this study was to determine the training and resources available in the digestive endoscopy services of different hospitals in Spain, in which sedation is performed by the nursing staff.

## 2. Materials and Methods

### 2.1. Design

A descriptive cross-sectional study was conducted in the Digestive Endoscopy Service at Toledo University Hospital (HUT) and various hospitals in central Spain. This study adhered to the Strengthening the Reporting of Observational Studies in Epidemiology (STROBE) checklist [20]. Data collection took place from June to September 2023.

### 2.2. Participants/Sample

The reference population was working nurses belonging to the digestive endoscopy service of the University Hospital of Toledo (HUT) and other hospitals in central Spain.

The inclusion criteria were as follows:Nurses from the endoscopy service of Toledo University Hospital and/or other hospitals in central Spain.Nurses with more than three months of experience in digestive endoscopy services.

The exclusion criteria were as follows:Health care professionals other than nurses.

The sampling was carried out with the GRANMO program for population estimation. Given that the current nursing staff of the Digestive Endoscopy service of the University Hospital of Toledo comprises 42 nursing professionals, a random sample of 38 individuals was sufficient for estimation, with a confidence of 95% and an accuracy of ±5 percentage units, a population percentage that will foreseeably be approximately 17% of sedation/analgesia, on the basis of a previous study by Campo et al. [21]. The percentage of replacements required is expected to be 10%. Because the reference population is small, it was expanded to hospitals close to the HUT to achieve a more representative sample. Convenience sampling was performed; all nurses were invited to participate in the HUT, and dissemination of information was carried out through online professional networks in nearby hospitals. In the HUT, the response rate was 97.6%, whereas in the other hospitals, the response rate could not be calculated since it was disseminated through professional networks.

### 2.3. Variables

The following independent sociodemographic variables were collected: age, sex, current position in the service, time of experience in the service, and previous knowledge of digestive endoscopy.

In turn, the following variables related to knowledge of sedation were also analyzed: previous experience in sedation, specific training in sedation, and the use of sedative drugs.

### 2.4. Procedure and Measuring Instruments

The data collection instrument used was the Nursing Sedation Survey, which is based on a sedation survey conducted in a national study in China [15]. This survey contains 19 items that, together, are used to evaluate the characteristics of the use of sedation, the composition of the staff, the equipment used, and the selection of medications, among others.

The HUT and various hospitals were contacted to present the study and request permission from the ethics committee. Once the necessary permits were obtained, an online questionnaire was designed with Microsoft Forms for the nursing staff, which was accompanied by an information sheet and informed consent. The nurses of the service were subsequently contacted to request their participation and collaboration in the study, and the link for completing the questionnaire was attached.

The data were collected in the second semester of 2023 to determine the training, skills, and resources needed for sedation in digestive endoscopies.

### 2.5. Statistical Analysis

The statistical analysis was performed using SPSS version 28, licensed to the University of Castilla-La Mancha. Qualitative variables are reported as counts (n) and percentages (%), while quantitative variables are presented as means (m) and standard deviations (SD). An inferential analysis was carried out to determine the differences between the variables depending on the hospital, and the proportions of the categorical variables were compared by means of chi-square tests for contingency tables. In the case of 2 × 2 tables, the chi-square statistic with Yates correction was used, and when any expected frequency was ≤ 5, Fisher’s exact test was applied. In addition, Pearson’s correlation was performed to study the relationships between age, time of experience, training in CPR, and previous experience in sedation. All hypothesis tests were bilateral. In all the statistical tests, statistically significant values were considered those whose confidence level was 95% (*p* < 0.05).

### 2.6. Ethical Considerations

This study received approval from the Ethics Committee of Clinical Research with Medicines at the “Complejo Hospitalario Universitario de Toledo” on 5 May 2023 (No. 1012). All participants reviewed the information sheet and gave their consent to take part in the study.

## 3. Results

A total of 75 nurses were included in the study. Among the total number of participants, 67 were women (89.3%), and the rest were men (10.7%). The average age of the participants was 37.29 ± 10.021 years (range 24–62 years) (Table 1).

With respect to training, 88% were general nurses, whereas a small percentage (9.3%) were specialized in sedation through ongoing training. With respect to service experience, the majority (48%) had been working for more than two years, which indicates a considerable accumulation of practical experience. However, 82.7% of the respondents had no previous experience with sedation before starting work in the endoscopy service. Despite this, 86.7% affirmed that sedation has been applied in their hospital.

Finally, life support training is also important: 56% of the respondents had training in basic life support, and almost half (49.3%) had training in advanced life support.

Table 2 shows the characteristics of the HUT nurses in comparison with the rest of the included hospitals.

As shown in Table 2, when the differences between nurses from different hospitals (HUTs and others) were analyzed, similarities were observed in terms of sex and profession. However, there were significant differences in basic life support training, where HUT hospitals had a lower proportion of nurses trained in this area than did other hospitals (39% vs. 76.5%, *p* = 0.001).

In the HUT, sedation by nurses is performed with Propofol only. In other hospitals, Propofol is also used in combination with other drugs such as Midazolam, Fentanyl, Sumifentanil, Remifentanil, and Ketamine (in order of frequency from most to least frequently used).

Table 3 shows the resources available in the different hospitals. In general, the resources available were extensive in all hospitals.

According to Table 3, there are significant differences in the availability of resources between hospitals. For example, the presence of nurses specializing in sedation through continuing education was much lower in “Other” hospitals (50%) than in HUTs (100%). Additionally, the availability of critical equipment such as respirators and EKG monitors varied significantly, which could affect patient safety during procedures.

According to Table 4, the most frequent complications perceived by endoscopy nurses during the administration of sedation in digestive endoscopy are hypoxemia, hypotension, and bradycardia, and less frequently aspiration and regurgitation; death during and after the examination and cardiac arrest being unusual or rare.

When performing the bivariate correlation (Table 5), a moderate association was found between age and previous experience, age, and training in basic life support. In addition, a moderate association was found between the time of experience and training in basic life support. An association was also found between previous experience in sedation services and training in basic and advanced life support. Finally, a strong association was found between having training in basic and advanced life support.

## 4. Discussion

In our study, in terms of the training of nurses, the vast majority were general nurses, and only a small percentage had specialized training through ongoing training. Most have been in service for more than two years. Very few had previous experience in sedation before starting to work in the service, although the vast majority currently applied sedation. Life support training was also critical. As for the HUT nurses, we can indicate that they all know where the defibrillators are located, since when they start their work, they are shown all the spaces in the unit and the location of the crash trolleys.

The data reflect a female predominance in the nursing field, which is consistent with global trends that show a higher proportion of women in the profession. The importance of sedation in endoscopic procedures, as highlighted by Tong et al. [1], stands out in its ability to improve patient tolerance of invasive procedures.

Most nurses had more than two years of service, suggesting that they had significant experience within the field. However, a high percentage lacked previous experience in sedation at the beginning of work in the endoscopy unit, highlighting a potential disconnect between years of service and specialization in sedation, which may influence the quality of support during procedures. In dealing with complications, it would be necessary to receive specialized training in sedation prior to incorporation into the digestive endoscopy service. In this context, training and competence in the sedation of nursing staff, as indicated, are crucial for ensuring the safety and effectiveness of the procedure [2,9], and the realization of robust educational programs and simulation practices is necessary.

In the context of sedation for gastrointestinal endoscopy, a review [3] revealed considerable variability in sedation recommendations among different guidelines, highlighting the need for standardization in training [14,15]. This finding motivates possible reflections on the opportunities for continuous training and specialization within hospitals, promoting uniform training standards aligned with the international guidelines of the WHO, where basic knowledge and real advanced performance should be required in sedations performed by non-anesthesiologists and considering the various legal frameworks and health care systems of each country. In addition, a more systematic study program and a regular education system that ranges from training to expert performance should be established to facilitate safe and successful endoscopic sedation [22]. This variability can impact patient safety and the efficiency of procedures, especially given the lack of a solid evidence base behind many of these practices.

In the United States, there is an established role for Certified Registered Nurse Anesthetists (CRNAs), who are highly trained and certified to provide anesthesia care. In contrast, in Europe, the advanced practice roles specializing in anesthesia are still in the process of development. In Spain, the competencies and functions of anesthesia nurses have been defined by the Spanish Association of Anesthesia, Resuscitation, and Pain Therapy Nursing (ASEEDAR-TD), which aims to ensure safe and effective sedation practices while promoting the necessary skills and knowledge among nursing staff.

Furthermore, while various European countries may have their own national guidelines and standards for sedation, there is ongoing dialogue within professional societies aimed at harmonizing training programs and best practices across borders. Initiatives such as collaborative training programs, workshops, and joint scientific meetings are important steps toward achieving greater standardization [23].

Likewise, the results reveal differences in life support training, both basic and advanced, among the different hospitals, which are generally deficient. The clinical practice guidelines of the Spanish Society of Digestive Endoscopy recommend that in digestive endoscopy units, all members should have the necessary skills to manage the airway and have a certificate of basic life support, which will be renewed every three years, and at least one member of the team must have certification in advanced life support or, failing that, have an anesthesiologist or an intensivist within a period of less than 5 min [13].

On the other hand, the differences observed between the hospitals in terms of training in basic and advanced life support may have an impact on the preparation of nurses to handle emergencies during sedation. At the national level, the guidelines of the Spanish Association of Gastroenterology (AEG) and other entities [11] recommend the review of procedures and continuous training of personnel to adapt to the changing needs of medical practice. This approach is essential to adequately respond to emerging challenges and ensure that all procedures are performed with the highest level of safety and efficiency possible. Conversely, the Spanish Association of Digestive Endoscopy (SEED) offers essential training for physicians and nurses involved in digestive endoscopy sedation by organizing courses at various hospitals across Spain throughout the year [13]. However, there are different ways of administering sedatives by nurses in endoscopic procedures that depend on the duration and complexity of the examination, as well as the personnel available in the unit [24]. In the case of the HUT, when performing a digestive endoscopy, there are two nurses in the room simultaneously: a nurse dedicated to sedation who is in charge of the administration of the sedative and the monitoring and continuous surveillance of the patient and another dedicated to the endoscopic procedure.

Technology also plays a crucial role in sedation training and practice. Xu et al. [25] conducted a randomized controlled trial with 154 patients who were divided into a group that used an artificial intelligence computer-assisted diagnosis (CAD) system to guide sedation (76 patients) and a control group (78 patients). Compared with the control group, the CAD group had significantly shorter emergency and recovery times after the endoscopic procedure. Patients in the CAD group also reported greater satisfaction with their level of sedation. The study concluded that the CAD system has great potential for improving anesthesia quality control and patient satisfaction during endoscopic procedures with sedation by allowing better adjustment of sedation levels. Tools such as virtual simulation and augmented reality are emerging as effective means to improve sedation training, providing realistic scenarios without risks for patients [26,27].

Most studies support the idea of the administration of sedation by nurses [28], but it is still necessary to establish common training programs and clinical practice guidelines [29].

The participating nurses reported the following complications as the most frequent: aspiration and regurgitation, hypoxemia, hypotension, and bradycardia, similar to other studies in which the most frequent complications in sedation in digestive endoscopies were measured [30,31,32].

Preparing and administering sedative medications, along with continuously monitoring vital signs and the patient’s reaction to sedation, are essential responsibilities [33,34]. Additionally, nurses play a vital role in the recovery phase, overseeing patients as they recover from sedation and assessing their readiness to resume normal activities [23,35,36,37,38].

The between-hospital comparisons revealed significant differences in the availability of specialized personnel and essential equipment, such as respirators and vital sign monitors. These discrepancies could have direct implications for patient safety and quality of care. The lack of uniformity in the availability of critical resources is an area of concern that needs to be addressed to ensure that all patients receive a safe and effective level of care, regardless of the hospital in which they are located. Technology and innovation [25,26] offer new opportunities to improve sedation training. The use of tools such as augmented reality and processor-assisted diagnostic systems can transform training, making it more effective and adaptive to the needs of staff, allowing high-fidelity simulations without risks to patients.

A positive correlation between age and time of experience is expected and reaffirms that accumulative experience is an important factor in the nursing profession. This aspect underscores the importance of aligning training opportunities with actual clinical responsibilities. Finally, the integration of practical experience and continuing education is essential to establish a framework of continuous training, and evidence-based evaluation is vital to maintain and improve skills in sedation, thus ensuring that patient care is managed with the utmost care and professionalism [16,39].

Nevertheless, sedation in digestive endoscopies has become a standard practice to ensure patient comfort and the effectiveness of the procedure. Adequate sedation training has been shown to improve the perceived quality of the procedure, reduce pain, and minimize untimely movements that could compromise the efficacy and safety of the procedure. All this contributes to greater patient satisfaction [13].

### Limitations

The first limitation lies in the cross-sectional design of the study, which, although useful for identifying correlations and trends at a specific time, does not allow us to establish causality or evaluate changes over time. This could limit the understanding of how training and experience in sedation evolve and affect clinical practice in the long term, so future studies on this topic would be helpful.

The second limitation is the difficulty in extrapolating the results. Given that the survey was distributed only among nurses from certain hospitals, the results could not be extrapolated to the entire Spanish health system that performs endoscopies. This may affect the generalizability of the findings, as different hospitals may have significant variations in sedation protocols, available resources, and levels of staff training. The non-response rate of the participating hospitals other than the HUT could not be calculated since the sampling was carried out through networks of online professionals, which may have favored the nurses who were the most motivated to participate in the study and respond to the questionnaire. However, the strength of this study is its representative sample of the HUT, and this is the first study in Spain in which satisfaction and training among nurses were analyzed together with the resources available to those who perform sedation in digestive endoscopy services.

## 5. Conclusions

In conclusion, the training of nurses who perform sedation in digestive endoscopy services can be improved, and the available resources are extensive. Given that there is a high proportion of personnel without specific training in sedation before starting work in digestive endoscopy services, it is crucial to implement targeted and continuous training programs to improve competencies in this area. These competencies are essential to guarantee the safety and effectiveness of the procedure.

Given the variability in resources and personnel training that exists between different hospitals, it seems important to establish standards at the institutional level for both resources and sedation practices. The inequality in education on basic and advanced life support, as well as in the availability of specialized equipment such as respirators and EKG monitors, suggests that establishing uniform standards could have a positive impact on both the quality of medical services and patient safety.

This study highlights how the combination of theoretical knowledge and clinical experience is essential to guarantee constant improvement in the safety of endoscopic procedures. The incorporation of advanced technology and the implementation of a more systematized and regulated approach to the training and practice of sedation will not only have advantages for healthcare professionals in terms of professional growth and job satisfaction but also guarantee the safety and well-being of the patient by following the best guidelines. The expertise and training of nursing staff responsible for administering sedation are vital for achieving safe and effective outcomes. Nurses should be well-trained in handling respiratory and cardiovascular emergencies that are commonly linked to sedation.

## Figures and Tables

**Table 1 healthcare-12-02087-t001:** Sociodemographic characteristics of the nurses.

	Frequency (n)	Percentage (%)
Sex		
Man	8	10.7%
Female	67	89.3%
Profession		
Nurse	66	88%
Nurse via EIR	2	2.7%
Nurse expert in sedation	7	9.3%
How long have you been in service?		
More than 3 months	7	9.3%
More than 6 months	14	18.7%
1–2 years	18	24%
More than 2 years	36	48%
Previous experience in sedation		
No	62	82.7%
Yes	13	17.3%
Has sedation been applied in your hospital?		
No	10	13.3%
Yes	65	86.7%
Has training in Basic Life Support		
No	33	44%
Yes	42	56%
Has training in Advanced Life Support		
No	38	50.7%
Yes	37	49.3%
Have you received previous training?		
No	46	61.3%
Yes	29	38.7%

**Table 2 healthcare-12-02087-t002:** Characteristics of the HUT nurses compared with those of other hospitals.

	HUT	Other	*p* Value
	n (%)	n (%)	
Sex			
Man	5 (12.2%)	3 (8.8%)	0.638
Female	36 (87.8%)	31 (91.2%)	
Profession			
Nurse	37 (90.2%)	29 (85.3%)	0.289
Nurse via RIN	0	2 (5.9%)	
Nurse expert in sedation	4 (9.8%)	3 (8.8%)	
How long have you been in service?			
More than 3 months	6 (14.6%)	1 (2.9%)	0.08
More than 6 months	10 (24.4%)	4 (11.8%)	
1–2 years	10 (24.4%)	8 (23.5%)	
More than 2 years	15 (36.6%)	21(61.8%)	
Previous experience in sedation			
No	37 (90.2%)	25 (73.5%)	0.057
Yes	4 (9.8%)	9 (26.5%)	
Has sedation been applied in your hospital?			
No	4 (9.8%)	6 (17.6%)	0.317
Yes	37 (90.2%)	28 (82.4%)	
Has training in Basic Life Support			
No	25 (61%)	8 (23.5%)	0.001
Yes	16 (39%)	26 (76.5%)	
Has training in Advanced Life Support			
No	25 (61%)	13 (38.2%)	0.05
Yes	16 (39%)	21 (49.3%)	
Have you received previous training?			
No	28 (68.3%)	18 (52.9%)	0.174
Yes	13 (31.7%)	16 (47.1%)	

Abbreviations: RIN, Resident Internal Nurse.

**Table 3 healthcare-12-02087-t003:** Resources available in the endoscopy sedation room.

	HUT	Other	*p* Value
	n (%)	n (%)	
Endoscopist			-
Yes	41 (100%)	34 (100%)	
No	0	0	
Endoscopy Nurse			0.002
Yes	41 (100%)	27 (90.7%)	
No	0	7 (20.6%)	
Sedation nurse			<0.001
Yes	41 (100%)	17 (50%)	
No	0	17 (50%)	
Anesthetist			0.804
Yes	11 (26.8%)	10 (29.4%)	
No	30 (73.2%)	24 (70.6%)	
TCAE			0.773
Yes	29 (70.7%)	23 (67.6%)	
No	12 (29.3%)	11 (32.4%)	
Respirator			<0.001
Yes	1 (2.4%)	23 (67.6%)	
No	40 (97.6%)	11 (32.4%)	
EKG monitor			0.115
Yes	41 (100%)	32 (94.1%)	
No	0	2 (5.9%)	
Take Oxygen			
Yes	41 (100%)	34 (100%)	
No	0	0	
Vacuum cleaner			0.269
Yes	41 (100%)	33 (97.1%)	
No	0	1 (2.9%)	
Mapleson system			<0.001
Yes	41 (100%)	22 (64.7%)	
No	0	12 (35.3%)	
Defibrillator			<0.001
Yes	41 (100%)	24 (70.6%)	
No	0	10 (29.4%)	
Stop trolley			<0.001
Yes	41 (100%)	24 (70.6%)	
No	0	10 (29.4%)	
Difficult airway material			<0.001
Yes	41 (100%)	23 (67.6%)	
No	0	11 (32.4%)	
Use Propofol			
Yes	41 (100%)	34 (100%)	
No	0	0	
Administer routine supplemental O_2_			0.058
No	0	1 (2.9%)	
Yes, 1–2 L/min nasal goggles	6 (14.6%)	12 (35.3%)	
Yes, nasal glasses 2–4 L/min	20 (48.8)	15 (44.1%)	
Yes, nasal glasses 4–6 L/min	15 (36.6%)	5 (14.7%)	
Yes, others	0	1 (2.9%)	
Monitor O_2_ Saturation			
Yes	41 (100%)	34 (100%)	
No	0	0	
Monitor EKG			<0.001
Yes	39 (95.1%)	18 (52.9%)	
No	2 (4.9%)	16 (47.1%)	
Monitor BP			<0.001
Yes	41 (100%)	19 (55.9%)	
No	0	15 (44.1%)	
Monitor RR			<0.001
Yes	41 (100%)	16 (47.1%)	
No	0	18 (52.9%)	
Monitors ETCO2			0.115
Yes	0	2 (5.9%)	
No	41 (100%)	32 (94.1%)	

Abbreviations: TCAE, technician in nursing auxiliary care; EKG, electrocardiogram; O_2_, oxygen; BP, blood pressure; RR, respiratory rate; ETCO2, carbon dioxide during expiration (end-tidal carbon dioxide).

**Table 4 healthcare-12-02087-t004:** Nurse’s perception of the main complications during sedation in endoscopy.

	HUT	Other	*p* Value
	n (%)	n (%)	
Death during exploration			
Never	41(100%)	27 (79.4%)	0.002
Rarely	0	7 (20.6%)	
Occasionally	0	0	
Frequently	0	0	
Very frequent	0	0	
Death after exploration			
Neer	38 (92.7%)	27 (79.4%)	0.092
Rarely	3 (7.3%)	7 (20.6%)	
Occasionally	0	0	
Frequently	0	0	
Very frequent	0	0	
Cardiac arrest during examination			
Neer	31 (75.6%)	20 (58.8%)	0.121
Rarely	10 (24.4%)	14 (41.2%)	
Occasionally	0	0	
Frequently	0	0	
Very frequent	0	0	
Aspiration and regurgitation during exploration			
Never	7 (17.1%)	4 (11.8%)	0.634
Rarely	18 (43.9%)	14 (41.2%)	
Occasionally	15 (36.6%)	16 (47.1%)	
Frequently	1 (2.4%)	0	
Very frequent	0	0	
Falling out of bed during examination			
Never	24 (58.5%)	25 (73.5%)	0.197
Rarely	16 (39%)	7 (20.6%)	
Occasionally	1 (2.4%)	2 (5.9%)	
Frequently	0	0	
Very frequent	0	0	
Unplanned hospitalization			
Never	4 (9.8%)	4 (11.8%)	0.691
Rarely	28 (68.3%)	20 (58.8%)	
Occasionally	9 (22%)	10 (29.4%)	
Frequently	0	0	
Very frequent	0	0	
Hypoxaemia			
Never	0	2 (5.9%)	0.038
Rarely	9 (22%)	15 (44.1%)	
Occasionally	24 (58.5%)	14 (41.2%)	
Frequently	8 (19.5%)	2 (5.9%)	
Very frequent	0	1 (2.9%)	
Hypotension			
Never	0	1 (2.9%)	<0.001
Rarely	2 (4.9%)	15 (44.1%)	
Occasionally	21 (51.2%)	16 (47.1%)	
Frequently	17 (41.5%)	2 (5.9%)	
Very frequent	1 (2.4%)	0	
Bradycardic			
Never	0	0	0.029
Rarely	3 (7.3%)	11 (32.4%)	
Occasionally	25 (61%)	18 (52.9%)	
Frequently	12 (29.3%)	5 (14.7%)	
Very frequent	1 (2.4%)	0	

**Table 5 healthcare-12-02087-t005:** Bivariate correlations between age, time of experience, training in CPR, and previous experience in sedation.

	Age	Experience Time	Basic Life Support Training	Advanced Life Support Training	Previous Experience in Sedation
Age	-	0.41 **	0.355 **	0.161	0.114
Experience time		-	0.305 **	0.185	0.091
Basic Life Support Training			-	0.66 **	0.264 *
Advanced Life SupportTraining				-	0.323 **
Previous experience in sedation					-

** *p* < 0.001; * *p* < 0.05.

## Data Availability

The data used to support the findings of this study are available from the corresponding author upon request.

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
