# Peer review of "Training and Resources Related to the Administration of Sedation by Nurses During Digestive Endoscopy: A Cross-Sectional Study"

_healthcare, 2024, doi:10.3390/healthcare12202087_

Round 1

Reviewer 1 Report

Comments and Suggestions for Authors

 Good cross sectional article.  I have a few questions

1. Did the nurses in HUT receive additional training? They knew 100% where the defibrillator is and about the mapleson system etc

2. Can you give a break down on complications - their total number and their incidence for both the groups.

3. Was propofol the only sedation being used?  No fentanyl, benadryl etc? 

4. Respirator - yes and no? Does it mean they were intubated for procedure yes and no?

5. In table 3 - what is the abbreviation for TA and FR?

Author Response

RESPOND TO REVIEWER

Ref: healthcare-3221818. “Training and resources related to the administration of sedation by nurses during digestive endoscopy: a cross-sectional study”.

We appreciate very much your constructive comments, helpful information and your time. We have considered all suggestions and incorporated them into the revised manuscript, and as a result, we believe our manuscript is stronger. Responses to his comments are written in bold type. We have highlighted in yellow the changes made to the manuscript.

REVIEWER 1:

  • Good cross-sectional article.  I have a few questions.

Thank you for your positive comments

  • 1. Did the nurses in HUT receive additional training? They knew 100% where the defibrillator is and about the mapleson system etc.

Thank you for your comment. As shown in table 3, only 31.7% of the HUT nurses received prior training in sedation when they started working in the service, 68.3% did not receive any prior training, but gained experience alongside more experienced colleagues during their working day.

The endoscopy service of the HUT has four defibrillators located next to the crash cart, properly located and signposted. All nurses know the location of the defibrillator. We have clarified this information in the discussion section.

All digestive endoscopy rooms are equipped with: EKG monitor, oxygen, aspirator and mapleson system, so 100% of the nurses are familiar with the equipment in the working rooms.

  • 2. Can you give a break down on complications - their total number and their incidence for both the groups.

Thank you for your comment. We have added the information in a new table. Table 4: Nurse's perception of the main complications during sedation in endoscopy.

  • 3. Was propofol the only sedation being used?  No fentanyl, benadryl etc? 

Thank you for your comment. We have added the information in the text.

In the Hospital Universitario de Toledo (HUT), sedation by nurses is performed with Propofol only. In other hospitals Propofol is also used in combination with other drugs such as: Midazolam, Fentanyl, Sumifentanil, Remifentanil and Ketamine (in order of frequency from most to least frequently used). We have added this information in resultados section.

  • 4. Respirator - yes and no? Does it mean they were intubated for procedure yes and no?

Thank you for your comment. There are two ventilators in the endoscopy department of the HUT which are used for more complex procedures or for emergency use by the anaesthesiologist.   The questions in the table refer to whether such resources exist in the endoscopy unit, but not whether nurses perform it.

The nurses did not intubate or use the ventilator for any of the patients sedated by the nurses either at the HUT or at another hospital.

  • 5. In table 3 - what is the abbreviation for TA and FR?

Thank you for your comment. We have modified the text.

Reviewer 2 Report

Comments and Suggestions for Authors

The shortage in anesthesiologist in many countries in addition to the responsible, short-time procedures force many countries to allow the nurses to administer anesthesia. It is expected that this policy will be more popular in the next era. There are many risk factors that must be fulfilled before this policy will be mandatory.

For myself, I believe Legalization will protect the nurse staff that in addition to scientific aspects with training of endoscopy nurse, the legal aspects had to be crystal-clear for everyone.

The idea looks attractive, however, I believe there still long way to go to be apply.

Few comments:

-This is a small number weak study done in a specialized tertiary hospital. You can not generalize these results in different hospitals with different supplements and staff variability. Hence, this is a strong drawback in the design. It is better to choose tertiary, medium-sized and small county hospitals to apply this type of research. Also, you need to design multinational study and not multi-center only.

-Before conducting such type of research, legalization is needed to protect your staff and the hospital itself. To my knowledge, this is not yet established.

-Is scientific training programs are standardized over Europe? I believe not yet.

-Is the basic and advanced life support will be enough to carry out the anesthesia by nurse?

-This issue underscore the location of endoscopy units nearby to the OR inside the hospitals for fast rescue help from OR; a situation that is not available in many hospitals all over the world. In addition, the alarm system should be quite tested daily.

Author Response

RESPOND TO REVIEWER

Ref: healthcare-3221818. “Training and resources related to the administration of sedation by nurses during digestive endoscopy: a cross-sectional study”.

We appreciate very much your constructive comments, helpful information and your time. We have considered all suggestions and incorporated them into the revised manuscript, and as a result, we believe our manuscript is stronger. Responses to his comments are written in bold type. We have highlighted in yellow the changes made to the manuscript.

REVIEWER 2:

-     This is a small number weak study done in a specialized tertiary hospital. You can not generalize these results in different hospitals with different supplements and staff variability. Hence, this is a strong drawback in the design. It is better to choose tertiary, medium-sized and small county hospitals to apply this type of research. Also, you need to design multinational study and not multi-center only.

We agree that the sample of our study is limited in the specialized hospital context in which it was carried out because the number of endoscopy nurses who apply sedation in endoscopy is limited. As you mention, this could affect the generalizability of the results to other hospitals so this information is shown in the limitations section.

However, our current study was designed as a first approximation to evaluate the training and resources available in the administration of sedation with nurses, with the intention of scaling up and replicating this study in other hospitals in future work. We have collected this information in the text.

-     Before conducting such type of research, legalization is needed to protect your staff and the hospital itself. To my knowledge, this is not yet established.

I appreciate your comment and understand the concern regarding the legalization of propofol administration by nurses in the hospital setting. In Spain, the regulation on the administration of propofol is clearly established. According to current regulations, propofol can be administered by nursing staff in sedation procedures, provided that they do so under the supervision of an anesthesiologist or a specialized physician. In the digestive endoscopy departments of several hospitals in Spain, this practice is protocolized and subject to strict guidelines that aim to ensure the safety of both staff and patients.

In our study, these regulatory frameworks were respected, and nurses participated in the sedation process following the hospital's guidelines and protocols. In fact, this study received the approval of the Head of the Endoscopy Department, and the hospital's ethics committee also supported the research. Therefore, the research we present is backed by the legislation and applicable regulations in the Spanish clinical setting. I am providing some references to support this:

  • Sociedad Española de Endoscopia Digestiva (SEED), Sociedad Española de Anestesiología, Reanimación y Terapéutica del Dolor (SEDAR). Sedation in Digestive Endoscopy: Clinical Practice Guidelines [Internet]. SEED; 2014 [cited 2024 Oct 9]. Available from: https://wseed.org/index.php/quienes-somos/guias-clinicas/227-sedacion-en-endoscopia-digestiva-guia-de-practica-clinica-de-la-sociedad-espanola-de-endoscopia-digestiva
  • Royal Decree 29/2006, of July 26, on guarantees and rational use of medicines and medical devices [Internet]. Official State Bulletin, no. 178, July 27, 2006 [cited 2024 Oct 9]. Available from: https://www.boe.es/buscar/act.php?id=BOE-A-2006-13554
  • Hospital General Universitario de Valencia. Protocolo de sedación en endoscopia digestiva [Internet]. Valencia: Hospital General Universitario de Valencia; 2007 [cited 2024 Oct 9]. Available from: https://chguv.san.gva.es/documents/10184/42181/ANFUEQUIR_Protocolo_Sedacion_Endoscopia_Digestivo_2007.pdf/456c5c29-475d-4d85-95b4-cdbf938063a0

Additionally, numerous studies support the administration of sedation by non-anesthesiologists (NAAP) or nurses (nurse-administered propofol sedation, NAPS). Many hospitals in different countries endorse this practice, with some having their own protocols. Most of these studies are related to endoscopic procedures in non-complex patients (ASA I and ASA II classifications) and critical care units. This practice is becoming increasingly common in non-operating room areas due to the growing number of diagnostic tests requiring sedation and the challenge anesthesiology services face in meeting this demand.

This increasing need has prompted more studies that demonstrate the safety of these sedation programs conducted by non-anesthetists and nurses. You can consult the following articles:

  • The review conducted by Minciullo et al. compiles various studies where this practice is performed. Minciullo, A., & Filomeno, L. (2024). Nurse-Administered Propofol Sedation Training Curricula and Propofol Administration in Digestive Endoscopy Procedures: A Scoping Review of the Literature. Gastroenterology nursing: the official journal of the Society of Gastroenterology Nurses and Associates, 47(1), 33–40.
  • Thomas SL, Rowles JS, editors. Nurse Practitioners and Nurse Anesthetists: The Evolution of the Global Roles. Cham: Springer International Publishing; 2023.
  • McKenzie P, Fang J, Davis J, Qiu Y, Zhang Y, Adler DG, et al. Safety of endoscopist-directed nurse-administered balanced propofol sedation in patients with severe systemic disease (ASA class III). Gastrointest Endosc. 2021;94(1):124–30.

In Spain, as mentioned in the introduction, the recommendations are based on the Spanish Association of Gastroenterology (AEG), the Spanish Society of Digestive Endoscopy (SEED), the European Society of Gastrointestinal Endoscopy (ESGE), the Asia-Pacific Consensus, the Spanish Society of Preventive Medicine, the European Panel on the Appropriateness of Gastrointestinal Endoscopy (EPAGE), and the guidelines from the Spanish Ministry of Health.

In this way, SEED in its clinical practice guidelines (Igea F, Casellas JA, González-Huix F, Gómez-Oliva C, Baudet JS, Cacho G, et al. Sedation for gastrointestinal endoscopy: clinical practice guidelines of the Spanish Society of Digestive Endoscopy. Rev Esp Enferm Dig. 2014;106(3):195–211) outlines the training and requirements necessary to perform it (type of procedure, level of sedation, etc.).

In this study conducted at the Hospital de la Fe in Valencia, Spain, a sedation protocol with Target Controlled Infusion systems was applied, which also demonstrated the safety of sedation. (Monsma-Muñoz M, Romero-García E, Montero-Sánchez F, Tevar-Yudego J, Silla-Aleixandre I, Pons-Beltrán V, Argente-Navarro MP. Retrospective observational study on safety of sedation for colonoscopies in ASA I and II patients performed by a nurse and under the supervision of anesthesiology. Rev Esp Anestesiol Reanim. 2022;69(6):319–25).

-      Is scientific training programs are standardized over Europe? I believe not yet.

I appreciate your comment regarding the standardization of scientific training programs in Europe. You are correct in pointing out that there is currently no uniform standard for training programs related to the administration of propofol sedation by nurses in digestive endoscopy services throughout Europe.

In the United States, there is an established role for Certified Registered Nurse Anesthetists (CRNAs), who are highly trained and certified to provide anesthesia care. In contrast, in Europe, the advanced practice roles specializing in anesthesia are still in the process of development. In Spain, the competencies and functions of anesthesia nurses have been defined by the Spanish Association of Anesthesia, Resuscitation, and Pain Therapy Nursing (ASEEDAR-TD), which aims to ensure safe and effective sedation practices while promoting the necessary skills and knowledge among nursing staff.

Furthermore, while various European countries may have their own national guidelines and standards for sedation, there is ongoing dialogue within professional societies aimed at harmonizing training programs and best practices across borders. Initiatives such as collaborative training programs, workshops, and joint scientific meetings are important steps toward achieving greater standardization. This information has been added to the discussion.

-     Is the basic and advanced life support will be enough to carry out the anesthesia by nurse?

We do not believe that this is sufficient, but currently in Spain the SEED (Sociedad Española de Endoscopia Digestiva) in its clinical practice guidelines indicates what is necessary to be able to perform sedation by a non-anaesthetist in this type of procedure. It can be consulted in: Igea F, Casellas JA, González-Huix F, Gómez-Oliva C, Baudet JS, Cacho G, et al. Sedación para endoscopia gastrointestinal: guías de práctica clínica de la Sociedad Española de Endoscopia Digestiva. Rev Esp Enferm Dig. 2014;106(3):195-211.

-     This issue underscore the location of endoscopy units nearby to the OR inside the hospitals for fast rescue help from OR; a situation that is not available in many hospitals all over the world. In addition, the alarm system should be quite tested daily.

Thank you for your comment. We think that in addition to the operating theatre, it is equally important to have a critical care unit nearby.

As has been observed in other studies and in our own, complications are minimal and are generally resolved satisfactorily by the nurses; however, in the case of the HUT, there is always an anaesthetist on duty, available to act in case of emergency.

Round 2

Reviewer 2 Report

Comments and Suggestions for Authors

It is OK.